# Environment, Endocrine Disruptors, and Fatty Liver Disease Associated with Metabolic Dysfunction (MASLD)

**DOI:** 10.3390/metabo14010071

**Published:** 2024-01-22

**Authors:** Antonella Mosca, Melania Manco, Maria Rita Braghini, Stefano Cianfarani, Giuseppe Maggiore, Anna Alisi, Andrea Vania

**Affiliations:** 1Hepatology and Liver Transplant Unit, Bambino Gesù Children’s Hospital, IRCCS, 00165 Rome, Italy; giuseppe.maggiore@opbg.net; 2Preventive and Predictive Medicine Unit, Bambino Gesù Children’s Hospital, IRCCS, 00165 Rome, Italy; melania.manco@opbg.net; 3Research Unit of Genetics of Complex Phenotypes, Bambino Gesù Children’s Hospital, IRCCS, 00165 Rome, Italy; mariarita.braghini@opbg.net (M.R.B.); anna.alisi@opbg.net (A.A.); 4Endocrinology and Diabetes Unit, Bambino Gesù Pediatric Hospital, 00165 Rome, Italy; stefano.cianfarani@opbg.net; 5Department of Systems Medicine, University of Rome Tor Vergata, 00133 Rome, Italy; 6Department of Women’s and Children’s Health, Karolinska Institutet, University Hospital, Solnavägen 1, Solna, 171 77 Stockholm, Sweden; 7Independent Researcher, 00162 Rome, Italy; andrea.vania57@gmail.com

**Keywords:** MASLD, EDCs, insulin-resistance, children

## Abstract

Ecological theories suggest that environmental factors significantly influence obesity risk and related syndemic morbidities, including metabolically abnormal obesity associated with nonalcoholic fatty liver disease (MASLD). These factors encompass anthropogenic influences and endocrine-disrupting chemicals (EDCs), synergistically interacting to induce metabolic discrepancies, notably in early life, and disrupt metabolic processes in adulthood. This review focuses on endocrine disruptors affecting a child’s MASLD risk, independent of their role as obesogens and thus regardless of their impact on adipogenesis. The liver plays a pivotal role in metabolic and detoxification processes, where various lipophilic endocrine-disrupting molecules accumulate in fatty liver parenchyma, exacerbating inflammation and functioning as new anthropogenics that perpetuate chronic low-grade inflammation, especially insulin resistance, crucial in the pathogenesis of MASLD.

## 1. Introduction

Ecological theories posit that a blend of environmental, social, and individual factors interplay in the onset of obesity and related syndemic morbidities [1]. Environmental factors contributing to the obesity epidemic include anthropogenic chemicals and endocrine disruptors (EDCs). The former are defined as “... *anthropogenic environments, their by-products, and/or the lifestyles they encourage, some of which may be harmful to human health*” and act by inducing a chronic condition of low-grade inflammation, also called “meta-inflammation”. The latter, defined as “*an exogenous substance or mixture that disrupts the functions of the endocrine system and consequently causes adverse effects in the organism or its progeny*” [2], can alter normal hormone levels by either inhibiting or stimulating hormone production and metabolism, and by modifying the way hormones are transported to the target tissues [3]. Developing organisms, from embryos to young adults, exhibit heightened sensitivity to EDCs due to the influence of hormones and growth factors on their development. The adverse effects of these EDCs may be more pronounced in developing organisms and occur at concentrations significantly lower than those deemed harmful in adults. Several factors contribute to this increased sensitivity in fetuses and newborns: immature protective mechanisms, such as DNA repair, an underdeveloped immune system, and the incomplete functionalities of detoxifying enzymes, the hepatic metabolism, and the blood–brain barrier. Additionally, developing organisms exhibit higher metabolic rates, and in some cases exposure to EDCs results in amplified toxicity [4]. Endocrine disruptors can alter genomic expression, potentially leading to epigenetic modifications that contribute to the development of various health issues. These effects include, but are not limited to, carcinogenic, neurotoxic, hepatotoxic, nephrotoxic, and immunotoxic outcomes. The liver, crucial for metabolizing many EDCs, also accumulates these lipophilic chemicals within its parenchyma, exacerbating inflammation and fibrosis in the presence of pre-existing steatosis [3,4]. Annual costs related to EDCs exposure have been estimated to approximate EUR 163 billion (above EUR 22 billion with a 95% probability, and above EUR 196 billion with a 25% probability).

In 2018, the European Commission asked the European Food Safety Authority (EFSA) and the European Chemicals Agency (ECHA) to develop a guidance document for the implementation of the scientific criteria for the determination of endocrine disrupting properties pursuant to EU Regulation no. 528/2012 on biocidal products.

This document was then published in the EFSA journal and reports the data relating to the presence of EDCs in waters of various countries around the world, raising the alarm on the extent to which their concentrations could be the basis of new diseases related to endocrine disruption, including NAFLD [5] (Table 1 and Figure 1).

**Table 1 metabolites-14-00071-t001:** Presence of EDCs in the waters of world countries.

Countries	Sample Type	Compounds	Concentration (ng/L or ng/g)	Reference
Taiwan (China)	River water	BPA	302	Tao et al., 2021 [6]
Kuwait	River water	DMP	16.9	Saeed et al., 2017 [7]
		DEP	524.8	
		DBP	899.0	
China	River water	NP	634.8	Wang et al., 2018 [8]
		BPA	1573.1	
		17 β-estradiol	23.9	
Vietnam	River water	PAEs	2.78–412.27	Quynh et al., 2019 [9]
		PBDEs	1.92–7.08	
Malaysia	River water	3,4,4-Trichlorocarbanilide	261.67	Aziz et al., 2014 [10]
		Methylparaben	4.93	
South Africa	River water	NP	2550	Farounbi et al., 2020 [11]
		Dichlorophenol	737	
		BPA	477	
Italy	River water	E1	28	Pignotti et al., 2017 [12]
		E2	39.7	
America	River water	E1	21.1–37.7	Sweeney et al., 2021 [13]
		E3	38.2–79.6	
		EE2	5.1–6.7	
Thailand	River water	E2	62.98 ± 5.03	Ocharoen et al., 2018 [14]
		EE2	35.53 ± 2.99	
		BPA	50.67 ± 4.19	
Brazil	River sediment	E1	187.39	Froehner et al., 2012 [15]
		DES	453.69	
		α-E2	21.36	
		β-E2	52.82	
		EE2	70.28	
		E3	34.68	

Bisphenol-S (BPS), diethyl phthalate (DEP), phthalates (PAEs), diethylstilbestrol (DES), di-(2-methoxyethyl)-phthalate (DMP), Bis(2-ethylhexyl) phthalate (DEHP), bisphenol A (BPA), 4-n-nonylphenol (NP), Polybrominated diphenyl ethers (PBDEs), ethinylestradiol (EE2), estrone (E1), 17-β-estradiol (E2) and estriol (E3).

**Figure 1 metabolites-14-00071-f001:**
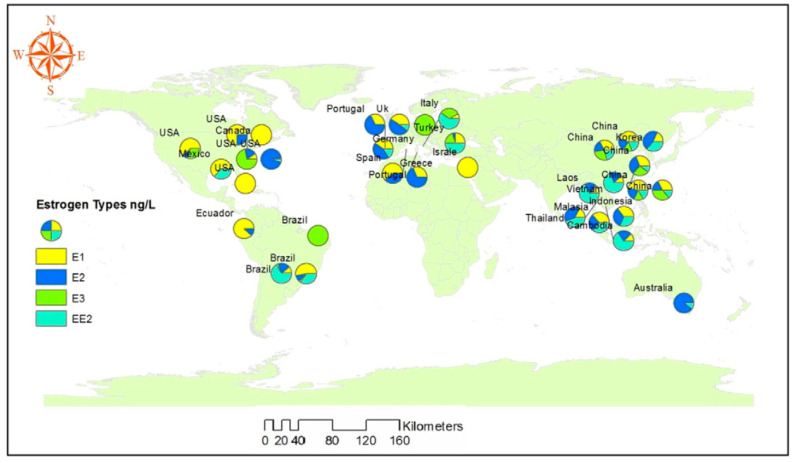
Worldwide distribution of natural and synthetic estrogens in surface waters and rivers [16].

## 2. Methods

Data sources and search strategy: A comprehensive literature search was conducted in PubMed databases (https://www.ncbi.nlm.nih.gov/ (accessed on 20 October 2023) using the following search terms: (“Endocrine disruptor(s)” OR any of the EDCs mentioned above) AND (“NAFLD”) or (“MAFLD”) or (MASLD”) AND (“obesity”) to locate all relevant published articles. Titles and abstracts were first scanned and full articles from potential eligible studies were independently reviewed by two researchers and were carefully examined for duplicates. Any discrepancies were resolved by consensus. The evaluation of included studies was conducted according to the PRISMA checklist criteria for completeness of systematic review results [17]. Data inclusion and exclusion: If all of the above criteria were met, studies were considered eligible for inclusion. We included only cohort studies and case–control studies, as well as animal model studies that assessed associations between IE and risk of NAFLD/MAFLD, and excluded non-original reports, review articles, conference abstracts, editorials, commentaries, experimental or pilot studies, case reports and case series, and studies without measures of exposure to EDs. The search was limited to studies published in English. Data extracted from each study include first author, country of origin, duration of studies, and sample size.

Sixty publications were found from 2013 to 2023, 22 of which were excluded because they were non-systematic reviews. Seven works that reported non-exclusively reported hepatic disease (PCOS, cognitive decline, immune disorders, bone diseases, hypothalamus pituitary adrenal axis alterations) were excluded. Twenty-four of the remaining studies were excluded because they had been conducted only on rats with NAFLD and obesity. Therefore, eight studies conducted on humans were considered, but only those with an adequate sample size were evaluated (Table 2).

## 3. Environmental EDCs and Sources of Exposure

EDCs encompass a broad spectrum of both natural and human-produced substances, originating from persistent organic pollutants (such as dioxins, benzo[a]pyrene and polychlorinated biphenyls (PCBs)), organochlorines (dichlorodiphenyltrichloroethane (DDT)), plasticizers (bisphenol A (BPA)), phthalates (di-2-ethylhexyl phthalate (DEHP)), organotins (tributyltin (polyfluoroalkyls and perfluorooctane sulfonate (PFOS))) and other pesticides (cypermethrin (CYP), atrazine (ATZ) and carbendazim). Approximately 1000 chemicals have been identified that meet the criteria for being classified as EDCs. These compounds are used across a diverse array of consumer products, including but not limited to food packaging, building materials, pesticides, clothing, detergents, plastics, and medical equipment. Some industrial process-related chemicals lead to unintentional contamination of food, water, and air. This exposes individuals through oral ingestion, dermal contact, inhalation, and even subcutaneous and intravenous routes (via medical equipment) [5].

The Stockholm Convention has identified a cluster of compounds termed the “dirty dozen”, which encompass pesticides, industrial chemicals, and their byproducts. This list includes aldrin, chlordane, toxaphene, mirex, hexachlorobenzene, heptachlor, furans, endrins, polychlorinated biphenyls (PCBs) and bisphenyl polybrominated (PBBs), polyfluoroalkyl substances (PFOS), dioxins, plasticizers (BPA, phthalates) pesticides (i.e., dichlorodiphenyltrichloroethane (DDT)), diethylstilbestrol (DES), and heavy metals (arsenic, cadmium and mercury) [26] (Table 3). Polycyclic aromatic hydrocarbons (PAHs) are generated through open combustion, natural filtration of oil or coal deposits, and incomplete combustion of various materials like coal, oil, gas, wood, waste, and tobacco [26]. PAHs can contaminate the environment by binding to airborne particles or adhering to foods cooked or roasted at high temperatures. Fluorinated chemicals (PFCs), extensively used in the manufacturing of cookware, clothing, food packaging, and firefighting materials, are widespread pollutants found in household environments. Due to their slow elimination from the body, they are classified as persistent organic pollutants (POPs), with a half-life ranging from approximately 2 to 8 years [26,27]. Bisphenol A (BPA) stands out as a notable EDC and is ranked third among the most significant pollutants according to the Environmental Protection Agency (EPA). BPA is an organic compound characterized by 2-phenolic rings connected by a methyl bridge with 2-methyl functional groups, classified in the phenol group. Due to its high strength as well as its hardness and transparency, BPA is widely used in industry as a material with which to produce phenolic resins, polyacrylates and polyesters and, above all, to produce plastics used in the manufacturing of everyday products [28]. Exposure to BPA happens via pollution, inhalation, and skin contact, but the most important route is from the consumption of contaminated foods, canned products, pre-packaged foods, packaged infant formula, baby bottles and containers used for food storage [26,27,28,29] (Table 3).

EDCs have been identified as compounds that mimic or block the action of estrogen receptors, androgens, and thyroid hormones, and, among them, BPA is considered a weak xenoestrogen [26].

While detectable in amniotic fluid, placental tissue, umbilical cord blood, and breast milk, determining the minimum toxic dose of BPA required to exert harmful effects remains challenging [9].

Despite its low lipophilicity and rapid degradation (with a half-life of 4–5 h), recent studies suggest a potential bioaccumulation in adipose tissue, due to BPA’s high affinity for fatty acids [46]. Corroborating this possibility, a further study demonstrated that, after a BPA-free diet for one month, a group of patients with NAFLD exhibited a reduction in circulating plasma BPA levels without a significant reduction of urinary levels compared with baseline [10]. This event could be a result of BPA release from adipose tissue, which would explain the reduction of BPA in plasma but not in urine after the BPA-free diet. Nonylphenols (NPs), also known as alkylphenols, are found in various products, including lubricants in oil additives, laundry detergents, emulsifiers, and solubilizers [47]. Alongside these chemicals, pesticides fall within the POPs category [48]. These substances exhibit a relatively slow degradation rate and high lipophilicity, resulting in their widespread presence in organic tissues, particularly adipose tissue. Being semi-volatile, POPs can disperse across long distances via air and ocean currents [49].

## 4. The Effects of Obesogens across the Lifespan

In 2009, Green and Blumberg highlighted that certain EDCs function as “obesogenic” substances (i.e., non-steroidal estrogens, organotins, parabens, phthalates, polychlorinated biphenyls and bisphenols), actively promoting obesity by altering endocrine homeostasis. They proposed that such substances modify homeostatic mechanisms critical to weight control, thereby increasing individuals’ susceptibility to obesity, even when they maintain a healthy diet and exercise routine [49]. The obesogenic hypothesis for EDCs rests on two pivotal points: first, there is an increased susceptibility to obesity, which starts in the prenatal period and continues across the first years of life and, secondly, this susceptibility to obesity is boosted by a specific subclass of EDCs, which alters the programming of human development, influencing weight control later in life [50]. Consequently, “obesogens” are functionally defined as chemicals that promote obesity by increasing the number of fat cells, and/or by enhancing fat accumulation in existing adipocytes. Recently, the domain of obesogens has broadened, acknowledging that certain EDCs exert organ-specific effects, promoting the onset and progression of morbidities that are syndemic with obesity. However, these organ-specific effects are not mediated by an increase in adiposity [51]. For instance, some EDCs induce, in both human and animal models, hepatic steatosis associated with metabolic dysfunction (MASLD) not only through epigenetic programming during early prenatal and postnatal life, akin to “classic” obesogens, but also by interfering directly with the functioning of hepatocytes [52]. There has been a recent transition to a new nomenclature for what was previously termed non-alcoholic fatty liver disease associated with metabolic syndrome. This new framework, relevant for children, integrates multifactorial considerations. However, before applying the diagnostic criteria for MASLD, it is crucial to rule out other causes of fatty liver disease, to avoid overlooking the possibility of a dual pathology [53]. The newly accepted nomenclature is hepatic steatosis disease (NAFLD) associated with at least one of the following conditions: excess adiposity (overweight/obesity or abdominal obesity), prediabetes or type 2 diabetes mellitus (T2DM), metabolic abnormalities (dyslipidemia, insulin resistance, alteration of the intestinal hepatic axis), as well as genetic predisposition and epigenetic events [54]. This new definition definitively closes the pathophysiological link of fatty liver with metabolic dysfunction and insulin resistance, reinforcing its role as cardiometabolic risk factors associated with MASLD. Hepatic MASLD is characterized by a broad spectrum of abnormalities ranging from simple steatosis to steatohepatitis associated with metabolic dysfunction (MASH), with the latter condition implicating lobular inflammation, swelling, and often fibrosis. To date, a multitude of genetic, epigenetic, and environmental MASLD modifiers have been reported, especially in the perinatal period and in the first years of life. However, diagnosis is often delayed due to its non-specific clinical manifestation, so that its discovery often occurs during routine examinations [53,54].

## 5. EDC and MASLD

### 5.1. Prenatal Life

The Developmental Origins of Health and Disease (DOHaD) theory emphasizes the existence of critical developmental windows during fetal stages, wherein environmental pressure can induce subtle changes in gene expression, tissue organization, or other levels of biological organization. These changes can lead to lasting dysfunction and heightened susceptibility to chronic disease [55]. Unlike birth defects or neonatal ailments, these dysfunctions often surface later in life, encompassing conditions such as obesity, NAFLD, and subsequently, MASLD [56]. Several studies have demonstrated that low birth weight (LBW) infants, born to mothers who experienced either excessive or deficient nutrition, are more prone to develop an increased susceptibility to various diseases in adult life. These include heart disease, high blood pressure, obesity, T2D, osteoporosis, dyslipidemia, impaired glucose metabolism, and NAFLD [54,55,56,57].

A recent prospective cohort study examined 253 children within a pediatric population and found association between prenatal exposures to ubiquitous EDCs (notably organochlorine pesticides, PBDEs, PFAS, and metals) and increased liver damage and/or hepatocellular apoptosis. Among 1108 children from 6 different countries, 253 (22.8%) exhibited a high risk of liver damage, indicated by elevated transaminases. All children in the study showed exposure to various EDCs, as follows: 3 organochlorine pesticides, 5 polychlorinated biphenyls, 2 polybrominated diphenyl ethers (PBDEs), 3 phenols, 4 parabens, 10 phthalates, 4 organophosphate pesticides, 5 perfluoroalkyl substances, and 9 metals. The prevalence of liver injury varied among countries, with the highest prevalence in children from Greece (80 out of 253, 31.6%) and the lowest in children from Lithuania (11 out of 253, 4.3%). Additionally, there was an escalated likelihood of liver injury corresponding to increased quartiles of exposure to organochlorinated pesticides (odds ratio (OR): 44.95, 1% credibility interval (CrI): 21.1–71.1), PBDEs (OR: 57.95, 1% CrI: 34.1–84.1), perfluoroalkyl substances (OR: 73.95, 1% CrI: 45.2–09.2) and metals (OR: 21.95, 1% CrI: 65.3–02.0) [25].

Several studies have shown that a range of chronic conditions, including obesity, chronic diseases, but also lifestyle habits, diet and physical activity, T2D and NAFLD, can be correlated with epigenetic modifications occurring in cells and tissues during development. These alterations affect impaired tissue development, arising from early environmental factors such as stress, pharmaceuticals, nutrition, and environmental chemicals. Importantly, the effects of diseases induced during development might not become immediately apparent but rather manifest later in life [57] (Figure 2).

Therefore, environmental monitoring data appear insufficient to predict population contamination levels and do not allow the detection of metabolite-related toxicities during fetal life. The experimental difficulties are also represented by the lack of knowledge of biological cause–effect relationships, which precludes the possibility of predicting the adverse effects of EDCs.

### 5.2. Young Adult Life

The liver stands as a crucial defense against harmful substances, employing complex and sophisticated mechanisms that include filtration, oxidation, and conjugation of chemical compounds. Enzyme systems such as cytochrome P450 (CYP) and UDP-glucuronosyltransferase (UGT) are responsible for eliminating more than 90% of these substances. While the complete metabolic pathways of toxins and the exact mechanisms by which EDCs disrupt liver function remain incompletely understood, it must be considered that their entry into the liver often coincides with food intake or contact with harmful materials. Consequently, the absorption and transport of EDCs often occurs alongside dietary lipids via the portal circulation. These compounds, characterized by high lipophilicity, swiftly diffuse across cell membranes, gaining expedited access to target sites. Upon reaching the liver, activation of nuclear receptors (NRs) by EDCs disrupts hormone signaling pathways and modulates the cytochrome P450 (CYP) system. This enzyme system comprises several isoforms, with CYP1A2, CYP2C9, CYP2C19, CYP2D6, CYP2E1 and CYP3A being the major players in toxin metabolism [52,58]. The influence of EDCs on this detoxification system remains a subject of debate. Studies have indicated that pesticides, parabens, phthalates, and BPA can reduce the catalytic efficiency of CYP450. These findings suggest that altered metabolism, coupled with prolonged exposure, significantly contributes to the bioaccumulation of these substances [59]. This intricate interaction can also convert some EDCs into more active metabolites. For instance, low molecular weight phthalates, such as di-(2-ethylhexyl) phthalate (DEHP), dimethyl phthalate (DMP), and dibutyl phthalate (DBP), can bio-transform into hydrolytic monoester metabolites with enhanced activity. An effective method to assess the impact of EDCs on liver function involves the use of traditional markers such as liver enzymes. Exposure to various POPs, including PCBs, occhlorinated dibenzodioxin (OCDD), and specific pesticides, has been linked to elevated bilirubin, ALT, and ALP levels. These elevations indicate a detrimental effect on liver function due to prolonged exposure to these pollutants in young adult populations [52,60,61,62].

It is therefore certain that individual genetic susceptibility, comorbidities and dietary habits can modify the impact of EDCs on the body. As examples on might look at the way a family history of T2DM or obesity may accentuate the diabetogenic effects of certain molecules; how a high-fat diet may increase the exposure to lipophilic EDCs; how EDCs modulating steroids’ action can give different results (depending on hormonal background, stage of body’s development, timing of exposure); and, finally, how trans-generational effects are possible. An important limitation to the interpretation of cross-sectional epidemiological studies is the lack of information on exposure to EDCs and possible confounding compounds. This does not allow one to establish causal links between exposure and effect.

## 6. Endocrine Disruptors and the Development of MASLD

Numerous EDCs (BPA, phthalates, PFOA and PFOS) are classified as “obesogenic”. The hypothesis put forth by Grun and Blumberg primarily focuses on the impacts of these exposures on adipocytes, leading to inflammation, oxidative stress, and effects on pancreatic β-cells. Consequently, this impact alters insulin secretion and contributes to the initiation and exacerbation of insulin resistance [63,64].

Endocrine disruptors possess the capacity to act as agonists or antagonists by binding to NRs. The liver harbors a vast array of NR [64]. The activation of classical NR activity typically involves ligand-mediated attachment to response elements situated in the promoter region of target genes. This is followed by binding of steroid receptor coactivator complexes (SRCs), recruiting additional regulators with diverse histone-modifying enzyme activities [65]. This extranuclear signaling leads to the activation of kinases and downstream signaling pathways that elicit biological responses irrespective of the nuclear positioning of NRs. Non-genomic signaling, characterized by its rapid action, occurs independently of RNA or protein synthesis [66].

Numerous NRs actively participate in non-genomic signaling. The most important of these are the steroid hormone receptors: estrogen receptors (ERα and ERβ), androgen receptors (ARs), and progesterone receptors (PRs). Additionally, other NRs, such as peroxisome proliferator-activated γ receptor (PPARγ), retinoid α X receptor (RXRα), thyroid α receptor isoforms (TRα), and retinoic receptor isoforms (RARα and RARγ), seem to employ similar signaling mechanisms. Retinoid X receptors (RXRα, RXRβ, and RXRγ), which include α, β, and γ receptors, are activated by peroxisome proliferators (PPARs), liver X receptors α and β (LXR), farnesoid X α receptor (FXRα), and thyroid receptors (TR), and have been implicated in the modulation of MASLD [25,26,67]. For instance, LXRs, when bound to oxysterols, activate lipogenic genetic programs such as FAS and SREBP1, leading to lipid accumulation in the liver. FXR responds to bile acids by inducing the expression of bile acid export genes and NR0B2, which in turn induces the expression of SREBP1. FXR can reduce fatty liver disease and insulin resistance. FXR agonists such as obeticholic acid are currently undergoing clinical trials for the treatment of NASH in humans [68].

Studies have shown that FXR inhibits the expression of fatty acid synthetase (FAS) and reduces the synthesis of fatty acids and triglycerides. The mechanism for this is the suppression of sterol regulatory element binding protein-1c (SREBP-1c) by FXR via SHP-mediated inhibition of co-activator recruitment to SREBP1c promoters [68]. Several studies in mice have shown that, when combined with other MASLD risk factors, EDCs typically induce the NAFLD phenotype in exposed rodents. A recent comprehensive review of 371 studies suggests that 123 unique environmental chemicals are associated with NAFLD in rodents, with pesticides accounting for the majority (44%), while PCBs and dioxins were the most potent, based on levels [69]. Two baseline studies have reported that perinatal exposure to BPA (50–100 μg/kg per day), combined with a high-fat diet (HFD) after weaning, induced more severe fatty liver disease and increased inflammation and mild fibrosis but that these effects were observed only in male subjects [69,70]. In vitro studies on HepG2 cells have highlighted BPA’s role in liver inflammation by triggering the release of proinflammatory cytokines (IL-8 and TNF-α). Acaroz et al. have reported increased levels of TNF-α, IL-6, and IL-1β, alongside decreased levels of IL-10, after low oral BPA doses, promoting the development of pro-inflammatory micro-environments and dose-dependent histopathological changes in rat livers [70].

These findings suggest that, apart from increasing hepatic lipid accumulation, exposure to EDCs may trigger inflammatory infiltration, exacerbating the development of NASH and MASLD [56,70]. The observed increase in hepatic lipid accumulation linked to BPA utilization might result from an imbalance in free fatty acid (FFA) absorption, synthesis, or β-oxidation and/or the export of triglycerides as very low density lipoprotein VLDL [56].

The impact of EDCs on long-term health, notably their capacity to alter the epigenome, has been extensively studied. These disruptions can induce lasting alterations in the epigenome which, due to the hereditary nature of epigenetic programs, can persist across many cell generations and throughout an individual’s life [71]. DNA methylation, the first molecular mechanism identified for the epigenetic regulation of gene expression, occurs through the enzymatic transfer of a methyl group to cytosine bases of DNA, resulting in the formation of 5-methylcytosine [70]. In animal models, several studies have explored how EDCs might interfere with DNA methylation in liver diseases [70]. In terms of pathologic findings, mice and rats treated with BPA showed dose-dependent dilation of liver tissue sinusoids, congestion, inflammation, and necrosis [71]. Prenatal BPA exposure has been associated with the metabolic health of offspring. In fact, such exposure has been shown to alter gene expression profiles and cause peripheral insulin resistance and hepatic lipotoxicity [72]. Rodent models further support the idea that gestational exposure to BPA can promote the development of NAFLD by perturbing the activity of nuclear transcription factors.

### Studies in Human Population with Obesity and MASLD

In the human population, the exposure to some EDCs, such as short-lived BPA, is nearly ubiquitous, with detectable urine levels found in up to 95% of individuals in the United States. In the case of dioxins and PCBs, characterized by a long-lasting nature, continuous exposure can lead to significant human intake through consumption of contaminated meat and water. While numerous cross-sectional epidemiological studies have been published, none conclusively establish causal relationships with disease [73].

Epidemiological studies have shown that exposure to BPA could have negative effects on NAFLD-related biomarkers, with higher levels of urinary BPA correlating with increased ALT levels [73]. Lang et al. report in their study that higher levels of BPA were associated with elevated GGTs (OR: 1.29; CI: 1.14–1.46) and abnormal ALP (OR: 1.48; 95% CI: 1.18–1.85) [74]. In the Korean National Environmental Health Survey, which included 3476 adult participants, the mean BPA concentration in the NAFLD group was significantly higher than in the non-NAFLD group (2.56 µg/L compared with 2.24 µg/L, *p =* 0.001). Similarly, a study involving adolescents from the NHANES in the United States demonstrated an increased risk of suspected NAFLD (ALT ≥30 U/L) in participants with significantly higher quartiles of BPA exposure [75].

PCBs have been subclassified into dioxin-like (DL) and non-dioxin-like (NDL) compounds. They persist in the environment and accumulate in soil, aquatic sediments, and species consuming these sources (like fish, cows, and other land animals) [76]. Despite the ban on their production and emission in 1979, human exposure to PCBs usually occurs through contaminated air, water, or food. These compounds accumulate in adipose tissue and are gradually released into the bloodstream [77]. The liver appears to be both a target and an effector organ for PCB-induced endocrine disruption. The onset of NAFLD results from an imbalance between lipid production and elimination, leading to excessive accumulation of hepatic lipids [75]. DLPCBs activate the aryl hydrocarbon receptor (AhR) and peroxisome proliferator-activated alpha and gamma receptors (PPARα/γ), exerting a multimodal effect on lipid accumulation and causing steatosis by disrupting hepatic lipid metabolism. Studies have shown that exposure to PCBs can induce NAFLD-related metabolic disorders, such as insulin resistance, obesity, and lipid metabolic dysfunction. PCBs also differentially regulate hepatic lipid metabolism and several related genes. Exposure to PCBs increases liver lipids and causes steatosis associated with toxic substances (mild small-droplet macro-vesicular steatosis) [78,79]. Positive associations have also been observed between PCB concentrations and NAFLD-related biomarkers, with most PCB congeners positively correlating with elevated ALT levels. As part of a study involving 4582 adults, Cave et al. report that 10.6% of participants showed an unexplained increase in ALT. Advanced age is significantly associated with total PCB levels in the top quartile of the participants aged 70 and older, with 71.7% showing elevated PCB levels compared with only 2.2% of those under 30 years of age [80]. Additionally, a study engaging 1108 mother–infant pairs from six countries found that prenatal exposure to PCBs is a potential risk factor for pediatric NAFLD and is associated with increased levels of CK-18, a marker of hepatocyte apoptosis [25].

Among the various EDCs implicated in the development of NAFLD, another notable compound is DEHP, a phthalate commonly used in a variety of products, including plasticizers in food wrappers and packaging, cosmetics, medical devices, and toys [81]. Phthalates can enter the human body through multiple routes, such as the skin, respiratory tract, and digestive tract [81]. These chemicals generally undergo rapid metabolism and are excreted within 24 to 48 h in the urine as glucuronide conjugates. DEHP appears to play a role in lipid metabolism and oxidative stress. Both DEHP and its active metabolite mono-(2-ethylhexyl) phthalate (MEPH) may contribute to hepatic triglyceride accumulation and worsen NAFLD [79,82]. MEHP’s impact on hepatocyte lipid accumulation might involve the inhibition of the Janus kinase 2/signal transducer and transcription activator 5 (JAK2/STAT5) pathway. This suggests that the regulation of STAT5 by MEPH could play a critical role in the activation of enzymes involved in fatty acid metabolism [83]. In addition, DEHP also mediates the deterioration of the antioxidant mechanism and induces oxidative stress. In a study involving 5800 Korean adults, the prevalence of NAFLD, as defined by the fatty liver index, was associated with elevated urinary levels of various phthalates, and higher quartiles of MEHP revealed a significantly higher risk (OR 1.39; 95% CI: 1.00–1.92) of NAFLD [83]. Another study, involving 102 subjects and which sought to examine the influence of MEP and MEHP on liver function, found that phthalate exposure may be associated with a statistically significant increase in serum ALT and AST levels, while urinary phthalate levels seemed to be correlated with increased serum triglycerides and decreased HDL cholesterol levels [84,85]. DEHP exposure also seems to impact the prevalence of NAFLD in adolescents. In a study involving 387 mother–child pairs in Australia, the authors found that prenatal exposure to DEHP was associated with a higher incidence of NAFLD at age 17 [86]. Additionally, research on 2308 adults with subclinical hypothyroidism (SCH) across NAFLD and non-NAFLD groups revealed that urine levels of phthalate metabolites were positively associated with NAFLD in those with SCH. This association could be explained by DEHP’s potential interference with thyroid function. DEHP has been noted for its antagonistic effect on thyroid receptors. Because thyroid hormones can activate the β TH receptor (a potential target in NAFLD therapy) and reduce fatty liver disease, the interference caused by DEHP might contribute to the induction or exacerbation of NAFLD [87].

Within the complex spectrum of NAFLD and MASLD and its pathophysiology, environmental exposure to chemicals such as EDCs present in industrial spaces represent a relevant trigger.

The effects of EDCs in NALFD occur through the activation of transcription factors that trigger imbalances between lipid influx/efflux in the liver, the promotion of mitochondrial dysfunction and the potentiation of major inflammatory responses in NASH progression. Therefore, the increasingly important role of EDCs in disease pathogenesis is undeniable, although evidence in human trials remains scarce.

## 7. Future Prospective

NAFLD associated with MASLD stands as the most prevalent liver ailment in children, encompassing a spectrum from simple steatosis to NASH, which can advance to cirrhosis [53]. It has emerged as a leading cause of liver transplantation in adulthood, following the advancements in hepatitis C virus (HCV) treatment [53,85,86]. The complexities of NAFLD’s comorbidities are multifaceted, with obesity, insulin resistance, and dietary factors serving as pivotal underlying pathophysiological mechanisms driving liver damage progression [53,87].

In addition to these established factors, environmental exposures to chemicals like EDCs have surfaced as potential contributors to increased susceptibility to MASLD. EDCs may actually modulate the transcription of hepatic lipid homeostasis gene expressions, thereby promoting MASLD [25]. Furthermore, exposure to EDCs during fetal and early childhood stages can impact the epigenome, causing alterations in DNA methylation and histone modifications. These changes influence hepatometabolic reprogramming by disrupting the expression of lipid pathway genes and interacting with steroid hormones NRs [87].

Given the pivotal role of the endocrine system in regulating development during prenatal, childhood, and puberty stages, exposure to EDCs poses a significant threat to fetal and infant development. The pervasive presence of EDCs in daily use consumer products heightens susceptibility among many children and adolescents to these adverse health effects [88].

Addressing environmental factors has become paramount in both the prevention and the treatment of MASLD. Implementing appropriate public health policies to mitigate chemical exposure before and during pregnancy, in the early years of life, and during sensitive developmental stages like puberty is crucial [89]. Hence, gaining deeper insights into the effects of EDCs on human health is imperative when devising future regulatory strategies aimed at averting exposure, and safeguarding children’s health.

Risk analysis can be improved by identifying the most exposed subjects and the predisposing factors to the adverse effects of EDCs and by extending the exposure analysis from blood and urine samples to those of the organs where the compounds accumulate (liver, adipose tissue) or which are relevant to the exposure perinatal (cord blood, placenta and breast milk). In addition to the direct measurement of EDCs, the development of clinical markers could facilitate identification and the prospective monitoring of exposed individuals, with the aim of establishing clear causal relationships between exposure and the development of MASLD.

## 8. Conclusions

While regulatory strategies are essential in terms of public health policies, the acquired awareness of the interplay between EDCs and the development of hepatic pathologies during critical life stages underscores the importance of considering, implementing, and integrating targeted preventive strategies and early clinical interventions into practice to preserve liver health in children and adolescents.

## Figures and Tables

**Figure 2 metabolites-14-00071-f002:**
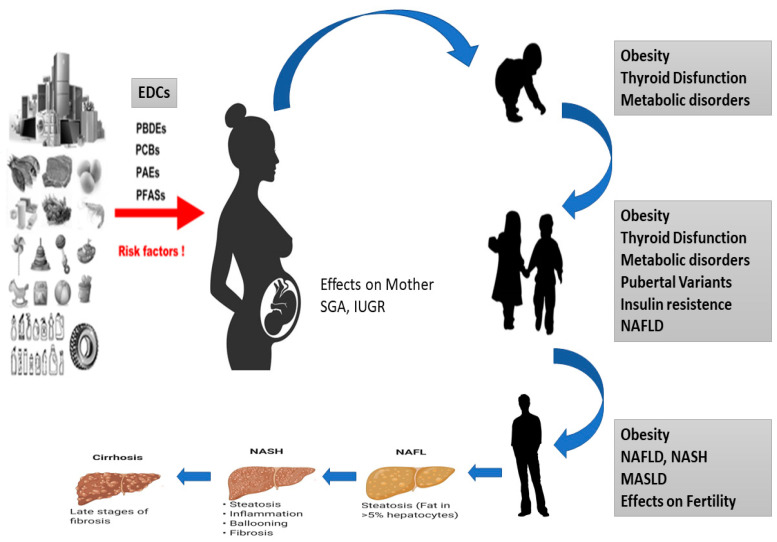
Effects of EDCs during development of MASLD in children. Diagram of how EDCs can affect the fetus, which can be born SGA or IUGR and is exposed to a series of epigenetic, genetic and environmental factors that can cause thyroid disease, obesity, metabolic disorders and NASH to develop in all stages of life.

**Table 2 metabolites-14-00071-t002:** EDCs and NAFLD development in human population.

Study	Authors	EDCs	Results
Cross-sectional cohort study	McCabe et al. [18]	PCBs	Twenty PCBs positively associated with elevated ALT levels (*p* < 0.05) in 436 adults
Cross-sectional study	Ke et al. [19]	Dioxins	Risk of fatty liver significantly increased in adults with higher BMI and higher serum PCDD/Fs (OR = 27.00, 95% CI = 4.47–229.58)
Cross-sectional study	Manikkam et al. [20]	PFAS	NASH significantly increased in 74 children, with increase in plasma concentrations of PFOS (OR: 3.32, 95% CI: 1.40–7.87), PFHxS (OR: 4.18, 95% CI: 1.64–10.7), and PFAS composite variable (OR: 4.89, 95% CI: 1.86–12.8).
Cross-sectional study	Ditzel et al. [21]	PFOA	Obesity and pathologies following DDT-induced epigenetic transgenerational inheritance of disease in 2216 adults
Cross-sectional study	Maranghi et al. [22]	BPA	Higher serum levels of BPA associated with higher grades of hepatic steatosis and AST, ALT, and GGT (*p* < 0.05) in women
Cross-sectional study	Boverhof et al. [23]	Phthalates	Correlations found between MEP concentration in urine and TAG serum levels (r2 = 0.33; *p* < 0.01), VAI (r2 = 0.41; *p* < 0.01), LAP (r2 = 0.32; *p* < 0.01), and TAG-to-HDL ratio (r2 = 0.40, *p* < 0.01) among 102 obese males
Cross-sectional study	Lei et al., 2021 [24]	3 EDCs metabolites (As, DiNP and PFOA)	In 5073 American adults the 3 EDCs metabolites significantly associated with MAFLD. ORs: 1.819 (95% CI: 1.224, 2.702), 1.959 (95% CI: 1.224, 3.136) and 2.148 (95% CI: 1.036, 4.456), respectively
Longitudinal population-based cohort studies	Midya et al., 2022 [25]	5 polychlorinated biphenyls, 2 polybrominated diphenyl ethers (PBDEs), 3 phenols, 4 parabens, 10 phthalates, 4 organophosphate pesticides, 5 perfluoroalkyl substances, and 9 metals.	A total of 1108 children, 253 of which (22.8%) classified as at high risk for liver injury. Increased ORs of liver injury per exposure-mixture quartile: 1.44 (95% CrI, 1.21–1.71) for organochlorine pesticides; 1.57 (95% CrI, 1.34–1.84) for PBDEs; 1.73 (95% CrI, 1.45–2.09) for PFAS; 2.21 (95% CrI, 1.65–3.02) for metals. Decreased ORs of liver injury associated with high-molecular-weight phthalates and phenols: 0.74 (95% CrI, 0.60–0.91) and 0.66 (95% CrI, 0.54–0.78), respectively.

ALT: alanine aminotransferase; AST: aspartate aminotransferase; BMI: body mass index; BPA: bisphenol A; CI: confidence interval; CrI: credible interval; GGT: gamma-glutamyl transferase; HDL: high-density lipoprotein; LAP: lipid accumulation product; MEHP: mono-(2-ethylhexyl) phthalate; MEP: monoethyl phthalate; NAFLD: non-alcoholic fatty liver disease; NASH: non-alcoholic steatohepatitis; OR: odds ratio; PCB: polychlorinated biphenyl; PCDD/Fs: polychlorinated dibenzo-p-dioxins and dibenzofurans; PFAS: perfluoroalkyl substances PFHxS: perfluorohexane sulfonic acid;; PFOA: perfluorooctanoic acid; PFOS: perfluorooctane sulfonate; TAG: triacylglyceride VAI: visceral adiposity index.

**Table 3 metabolites-14-00071-t003:** Main EDCs implicated in hepatic injury.

EDCs	Name	Molecular Targets	Study Model
**Solvents/lubricants**	Bisphenyl polychlorinates (PCBs)	Corticosterone levels, liver fibrosis	Mice, male [30]
	Bisphenyl polybrominated (PBBs)	Fecundit cells, liver steatosis	Mice [31]
	Per- and polyfluoroalkyl substances (PFOS)	Liver injury, PFASs	mice [32]
	Dioxins	Cell-mediated immunity, liver injury	Human, mice [33]
**Plasticizers**	Bisphenol A (BPA)	Fertility, liver steatosis	Mice, male female [34]
	Phthalates	Insulin resistance and type II diabetes, overweight and obesity, liver steatosis	Male, female and children [35]
	Di(2-ethylhexyl) phthalate (DEHP)	Liver injury	Mice [36]
**Pesticides**	Cypermethrin (CYP), atrazine (ATZ)	liver injury, growth parameters	Mice [37]
	Dichlorophenyltrichloroethane (DDT)	neonatal body weight, liver damage	Mice, male, children [38]
	Permethrin	Dopamine transport, fatty liver	Mice, human [39]
**Drugs**	Diethylstilbestrol (DES)	Expression of PDGF receptor, neonatal body weight, fatty liver	male and female/mice [40]
**Heavy Metals**	Arsenic	apoptotic index, liver injury	Mice [41,42]
	Cadmium	Expression of metallothionein, pS2/TFF1, liver steatosis	Mice [43,44,45]
	Mercury	Growth, hepatic steatosis	Mice, human [44]

## Data Availability

All data presented and reported in this article come from the scientific articles listed in the References section.

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
