# Peer review of "Environment, Endocrine Disruptors, and Fatty Liver Disease Associated with Metabolic Dysfunction (MASLD)"

_metabolites, 2024, doi:10.3390/metabo14010071_

Round 1
Reviewer 1 Report
Comments and Suggestions for Authors
See attached file

See attached file
Reviewer 2 Report
Comments and Suggestions for Authors
Mosca et al. presented the review “Environment, Endocrine Disruptors, and Fatty Liver Disease 2 Associated with Metabolic Dysfunction (MASLD)”. After a brief description of the Endocrine Disruptors Chemicals (EDCs), the authors described the role of these chemicals in the development of MASLD, reporting an in-depth series of studies relating to the molecular mechanisms underlying the pathology.
I have a suggestion for authors:
To offer a quick and effective analysis to readers, the authors could create a table (Table 2) reporting for each EDC reported in the review the mechanisms of action, the molecular targets, the study model, and the citations.
Other minor comments:
-Lines 93-95, the sentence is repetitive, interrupts the discussion on BPA, and can be eliminated.
-Line 106, Nonylphenols (NPs) are EDCs? Why are they not listed in Table 1?
-Table 1: The table reports “Main EDCs implicated in hepatic steatosis”. Please complete Table 1 with all the substances indicated in paragraph 2, if relevant, and report the references by creating a column on the right reporting the citations.
-Genistein is referred to as EDC/obesogenic compound, however many studies indicate that it does not cause hepatic steatosis.
Round 2
Reviewer 1 Report
Comments and Suggestions for Authors
The authors have successfully completed my comments.
Author Response
Thank you very much for your comments which significantly improved the manuscript
Reviewer 2 Report
Comments and Suggestions for Authors
The authors have provided a new version of the review, answering the comments of the reviewers.
I am still perplexed by the fact that the authors focused on EDC substances that would induce hepatic steatosis (Table 3: Main EDCs implicated in hepatic steatosis.), while the references provided deal with other aspects completely detached from the main theme of the review. If it is the authors' intent, the title of Table 3 should be changed in line with the content of paragraph 2. Otherwise, Table 3 should be rearranged.
Furthermore, for all the substances listed in Table 3, authors must briefly comment on the text, with the corresponding references.
Author Response
Thanks for your comments.
We have modified Table 3.
EDCs that are not purely hepatic have been eliminated from the table;
the title of the table has been changed and the hepatotoxic EDCs reported in the table have been inserted into the text.
The bibliography was then updated.